# HIFI-VC: High Quality ASR-based Voice Conversion

*Anton Kashkin, Ivan Karpukhin,[*] Svyatoslav Shishkin*

Tinkoff

a.kashkin@tinkoff.ru, i.a.karpukhin@tinkoff.ru, s.shishkin@tinkoff.ru

## Abstract

The goal of voice conversion is to convert the input voice to match the target speaker's voice while keeping text and prosody intact. Voice conversion is usually used in entertainment and speaking-aid systems, as well as applied for speech data generation and augmentation. The development of any-to-any voice conversion systems, which are capable of generating voices unseen during training, is of particular interest to both researchers and the industry. Despite recent progress, any-to-any conversion quality is still inferior to natural speech.

In this work, we propose a new any-to-any voice conversion pipeline. To the best of our knowledge, it is the first use of an ASR encoder with a GAN training objective in the voice conversion system. We also implement a joint conditional decoder-vocoder model, which simplifies training and improves performance. According to multiple subjective and objective evaluations, our method outperforms modern systems in terms of voice quality, similarity, and consistency.

**Index Terms**: any-to-any voice conversion, speech synthesis

## 1. Introduction

Speech synthesis aims at generating waveforms containing voices with desired properties [1, 2]. The two main approaches to speech synthesis are text-to-speech (TTS) and voice conversion (VC). In TTS, an algorithm predicts a voice waveform based on the provided text. Sometimes extra information is available, such as emotion, tempo, or the desired target voice sample [3]. In voice conversion, an algorithm converts the voice of one speaker to the voice of another speaker without affecting textual content and prosody. A simple VC algorithm can be implemented by combining an automatic speech recognition (ASR) system with TTS. However, the textual bottleneck of this approach leads to losing important information about prosody. To tackle this problem, special VC algorithms were proposed by several authors for use in entertainment [4], speaking-aid systems [5], data augmentation [6] and anonymization [7]. Despite recent improvements, the quality of converted speech is still inferior to natural speech, and future development is required.

Voice conversion methods can be characterized by their level of complexity. Basic approaches convert the voice of one or multiple predefined speakers to the voice of a single target speaker [8]. There are also *many-to-many* methods, that are capable of converting voices in a closed set of speakers, provided during training [9, 10]. Finally, the most general approach to VC aims at converting arbitrary voices either seen or unseen during training [11, 12]. This approach is usually called *any-to-any* or *zero-shot* voice conversion. In this work, we address the

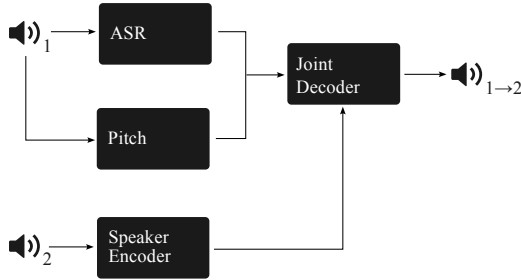

Figure 1: *HiFi-VC inference pipeline. We use pretrained ASR bottleneck features and a pitch tracker for encoding. Decoder and vocoder are joined into a single model with an additional condition on the output of a speaker encoder.*

most general form of voice conversion, namely any-to-any VC, where we show superior quality compared to popular methods.

Most recent works perform voice conversion in three steps [13, 12]. First, they extract content and speaker features from an input sample, then they generate a spectrogram and finally convert it to a waveform using a vocoder. In contrast to these works, we simplify architecture by joining the decoder and vocoder into a single model. Several works use neural waveform encoders, including ASR, and decoders [9, 11, 14]. Some approaches exploit ideas from generative-adversarial networks (GAN) [11, 15]. While both ASR and GAN improve conversion quality, there were no attempts to join these methods in a single pipeline.

In this work, we propose a new high-quality any-to-any voice conversion system. The proposed model structure is presented in Figure 1. Contributions of this paper can be summarized as follows:

1. We for the first time combine ideas from ASR-based content encoding with a GAN training objective to achieve high-quality any-to-any voice conversion. Ablation studies demonstrate the importance of each component for output speech quality.

2. The proposed conditional HiFi-GAN architecture is capable of directly predicting a waveform from intermediate features. In particular, we adapt HiFi-GAN [16] vocoder for a general decoding task.

3. We compare the proposed method with modern approaches using subjective and objective evaluations. According to our experiments, the proposed method improves voice quality, similarity, and consistency. The inference code and voice samples are publicly released[1].

---

[*]Corresponding author.

[1]https://github.com/tinkoff-ai/hifi_vc

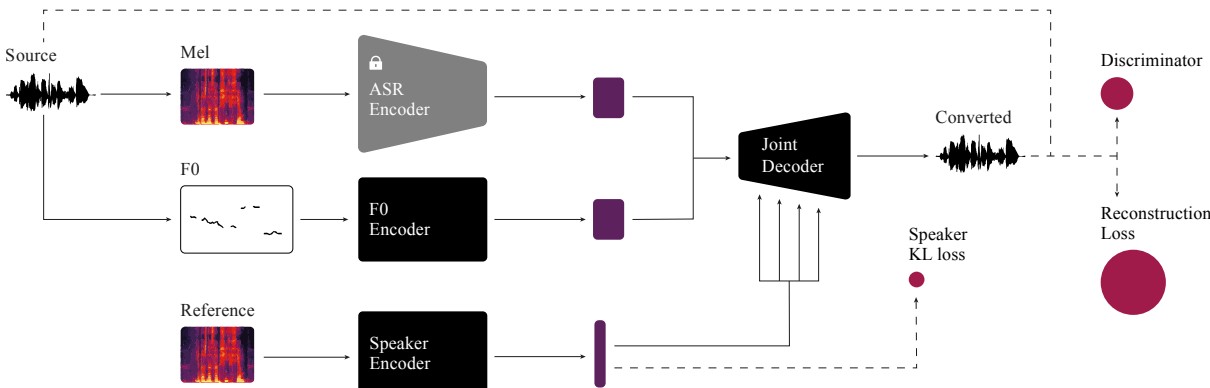

Figure 2: *HiFi-VC training pipeline. Output waveform combines linguistic information and prosody from the source sample with reference timbre. The ASR model encodes the linguistic content, while the pitch encoder provides prosody information. The pretrained ASR model used in the content encoder is frozen during training.*

## 2. Related Work

Given a source speech sample, VC methods try to extract speaker-agnostic content information. Some methods implement a bottleneck layer for information decoupling [13]. Additional normalization layers can be applied to further improve speaker and content features disentanglement [17]. Some works explicitly force disentanglement by either minimizing mutual information [12], by matching representations of the original and converted speech samples [11] or by using discriminators [18]. The corresponding encoders have little knowledge about the language and thus miss important content information.

Several works apply an ASR model for content encoding [10, 19]. Some methods use phonemes probabilities on the output of ASR [19], while others utilize bottleneck features [10, 9]. When it comes to the latter, it was empirically observed that bottleneck ASR features contain little information about the source speaker. Another use of ASR is to design a training objective that minimizes the loss of linguistic information during conversion [15]. One drawback of ASR-based coding is ASR's practical inability to encode prosody features. To handle this problem, most methods directly extract fundamental frequency (F0) from the source sample [10, 15]. Speaker information is usually eliminated from F0 by applying normalization [12]. The encoder part of our method includes ASR and F0 extractor similar to TTS Skins [10]. F0 is additionally preprocessed by a trainable network similar to that in BNE-Seq2seqMoL [20].

Given content and speaker features, most methods decompose prediction into two steps: decoding and vocoding [13, 12]. The goal of the decoder is to predict the Mel spectrogram of the output signal. The vocoder, on the other hand, converts the predicted spectrogram into an output waveform. The decoder can be implemented via RNN [13], Transformer [19], or a fully convolutional architecture [18]. Popular vocoders include Griffin-Lim [21], WORLD [22] and neural network models such as WaveGlow [23] and HiFi-GAN [16]. To handle Mel spectrogram prediction errors, an extra vocoder fine-tuning step is required after decoder training.

In this work, we propose a joint decoder-vocoder HiFi-GAN module with additional speaker conditioning as shown in Figure 1. Unlike previous works based on ASR and GAN [9], we avoid the intermediate Mel spectrogram prediction step

and directly produce a waveform from encoded features. Some previous works, such as VCRSS [9] and NVC-Net [11], also avoid using a separate vocoder. Unlike VCRSS, we use a GAN decoder for better waveform prediction. Our approach is also different from NVC-Net, as we use the ASR encoder and a different GAN architecture.

## 3. Proposed Method

Our model is based on an encoder-decoder architecture with an additional speaker encoder, as shown in Figure 2. During training, the reference record equals the source record, and the model learns to separate content information from the voice. In inference time, we can use an arbitrary reference sample for encoding the target voice. The architecture and training details are described below.

### 3.1. Model Architecture

**ASR encoder.** The goal of ASR encoder is to extract speaker-agnostic content information from the source voice sample. We do this by using bottleneck features from an automatic speech recognition (ASR) model. In particular, we apply the Conformer ASR [24] pretrained by NVIDIA and available on the official website[2]. The usage of the pretrained ASR allows us to largely speed up training and improve generalization as ASR representations are independent of the particular VC dataset.

**F0 encoder.** As ASR is trained to extract linguistic information, it is not very accurate at capturing prosody. To overcome this limitation, we add a fundamental frequency (F0) predictor similar to the BNE-Seq2seqMoL approach [20]. We also extract the boolean vocalization feature, which indicates regions where F0 can't be estimated. Our F0 predictor consists of the WORLD extractor [22] and a fully-convolutional network with 3 layers and instance normalization [25], which produces 256-dimensional vectors and downsamples input signal four times to match the ASR frequency. The goal of instance normalization is to exclude speaker information from F0, while a trainable subnetwork provides more flexibility in F0 coding.

---

[2] https://catalog.ngc.nvidia.com/orgs/nvidia/ teams/nemo/models/stt_en_conformer_ctc_large_ls

Table 1: *Many-to-many voice quality and similarity mean opinion scores (MOS) along with objective metrics for the proposed method and baselines. "F" and "M" correspond to different gender combinations of source and reference voices. Word error rate (WER) and character error rate (CER) are reported in percentages. Standard deviation (STD) of the voice quality in all studies is less than 0.19. STD of similarity is less than 0.27. Ground truth score is obtained using original records from the dataset.*

| Model | Voice Quality ↑ | | | | | Similarity ↑ | | | | | Objective Metrics | | |
|---|---|---|---|---|---|---|---|---|---|---|---|---|---|
| | F2F | F2M | M2M | M2F | Mean | F2F | F2M | M2M | M2F | Mean | WER↓ | CER↓ | PCC↑ |
| Ground Truth | 4.30 | N/A | 4.35 | N/A | 4.33 | 4.37 | N/A | 4.44 | N/A | 4.40 | 0 | 0 | 1 |
| AutoVC[13] | 2.22 | 2.14 | 2.27 | 2.15 | 2.20 | 2.26 | 2.42 | 2.09 | 2.49 | 2.32 | 85.1 | 58.1 | 0.22 |
| VQMIVC[12] | 3.93 | 3.69 | 3.74 | 3.78 | 3.78 | 2.97 | 3.10 | 3.19 | 2.97 | 3.06 | 32.5 | 16.9 | 0.51 |
| NVC-Net[11] | 3.73 | 3.17 | 3.71 | 3.35 | 3.49 | 3.91 | 3.79 | 3.83 | 3.71 | 3.81 | 37.9 | 21.4 | 0.42 |
| PPG-VC[20] | 3.64 | 3.72 | 3.84 | 3.72 | 3.73 | 1.60 | 2.38 | 2.32 | 1.51 | 1.95 | 16.7 | 8.0 | 0.38 |
| HiFi-VC-no-ASR | 2.02 | 2.01 | 2.04 | 1.95 | 2.00 | 3.90 | 4.01 | 3.62 | 4.03 | 3.89 | 92.8 | 64.2 | 0.55 |
| HiFi-VC-no-F0 | **4.33** | **4.37** | **4.36** | **4.29** | **4.34** | **4.33** | 4.23 | 4.23 | **4.26** | **4.26** | **12.3** | **5.3** | 0.33 |
| HiFi-VC | 4.20 | 4.19 | 4.17 | 4.18 | 4.18 | 4.22 | **4.31** | **4.31** | 4.13 | **4.26** | 13.1 | 5.4 | **0.61** |

**Speaker encoder**. We implement any-to-any voice conversion by using a speaker encoder network. The speaker encoder predicts the distribution of speaker feature vectors from an audio sample using a 5-layer residual fully-connected network. The multivariate normal distribution is defined by the mean vector and the diagonal covariance matrix. During training, speaker features are obtained via sampling, while in testing mean is used. The speaker encoder is similar to the encoder part of VAE [26] and is trained along with other modules in a single pipeline.

**Decoder**. The goal of the vocoder is to convert a Mel spectrogram into a waveform. Some previous works utilize HiFi-GAN [16] as a vocoder model. Our novel decoding approach extends previous GAN-based methods, as shown in Figure 3. While previous HiFi GAN-based approaches use different networks for the decoder and vocoder, we combine both into a single model. We implement a conditional HiFi-GAN, with conditions obtained from the speaker encoder. In particular, ASR bottleneck and F0 features are directly served as generator input. Speaker embeddings are linearly projected to match the dimensions of HiFi-GAN upsampling blocks and then added to their inputs. In general, our approach simplifies the prediction pipeline and doesn't need a vocoder fine-tuning step after decoder training. Furthermore, speaker embedding controls generation at all levels, including the waveform prediction step.

**Discriminator**. We use the same discriminator architecture as in HiFi-GAN [16].

### 3.2. Training Objectives

During training, we freeze the ASR model and simultaneously optimize the parameters of the F0 encoder, speaker encoder, decoder network, and GAN discriminator. We do this by combining modified HiFi-GAN losses [16] with speaker encoder regularization loss from NVC-Net [11].

In our method, we do not use intermediate Mel-spectrogram representation. In contrast to the original HiFi-GAN approach, we compute reconstruction loss between spectrograms of the source and predicted voice samples. Suppose $s$ is the predicted waveform, $x$ is the natural one and $M$ is a Mel-spectrogram transform. Here, reconstruction loss is defined as

$$\mathcal{L}_{Rec}(s, x) = ||M(s) - M(x)||_1. \quad (1)$$

HiFi-GAN applies adversarial loss to make the predicted waveform similar to the natural one. Suppose $D$ is a discrimi-

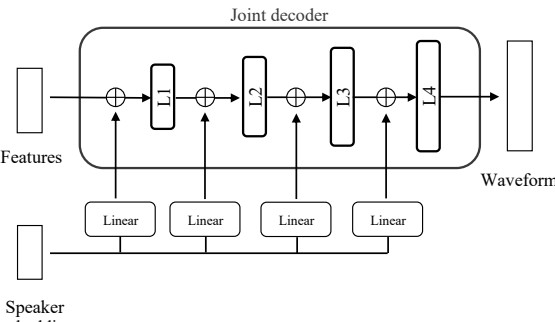

Joint decoder

Features

Speaker embedding

Waveform

Figure 3: *The conditional HiFi-GAN decoder architecture. Speaker embedding is projected using linear layers, repeated along time dimension, and then added to generator layers inputs.*

nator network. Then, the loss used for predictor optimization is defined as

$$\mathcal{L}_{AdvP}(s) = (D(s) - 1)^2, \quad (2)$$

and the loss for discriminator optimization is defined as

$$\mathcal{L}_{AdvD}(s, x) = (D(x) - 1)^2 + (D(s))^2. \quad (3)$$

GAN training is stabilized using feature matching loss. Suppose an output of the $i$-th discriminator layer is an $N_i$-dimensional vector $D_i$. In this case, feature matching loss is computed as

$$\mathcal{L}_{FM}(s, x) = \sum_{i=1}^{T} \frac{1}{N_i} ||D_i(x) - D_i(s)||_1. \quad (4)$$

In addition to the above, there is also speaker embedding regularization loss, which prevents a distribution collapse. Suppose, there is a reference record $r$. If we define the mean and diagonal covariance matrix predicted by the speaker encoder as $\mu(r)$ and $\Sigma(r)$, then regularization loss would be defined as

$$\mathcal{L}_{Spk}(r) = D_{KL}(\mathcal{N}(x; \mu(r), \Sigma(r))||\mathcal{N}(x; 0, I)). \quad (5)$$

The final loss for predictor is a weighted sum of the losses described above:

$$\mathcal{L}_P(s, r, x) = \lambda_{Rec}\mathcal{L}_{Rec}(s, x) + \lambda_{AdvP}\mathcal{L}_{AdvP}(s) + \\ \lambda_{FM}\mathcal{L}_{FM}(s, x) + \lambda_{Spk}\mathcal{L}_{Spk}(r). \quad (6)$$

Table 2: *Any-to-any voice quality and similarity mean opinion scores (MOS) along with objective metrics for the proposed method and baselines. "F" and "M" correspond to different gender combinations of source and reference voices. Word error rate (WER) and character error rate (CER) are reported in percentages. The standard deviation (STD) of the voice quality in all studies is less than 0.23. STD of similarity is less than 0.29. The ground truth score is obtained using original records from the dataset.*

| Model | Voice Quality ↑ | | | | | Similarity ↑ | | | | | Objective Metrics | | |
|---|---|---|---|---|---|---|---|---|---|---|---|---|---|
| | F2F | F2M | M2M | M2F | Mean | F2F | F2M | M2M | M2F | Mean | WER↓ | CER↓ | PCC↑ |
| Ground Truth | 4.27 | N/A | 4.47 | N/A | 4.37 | 4.39 | N/A | 4.17 | N/A | 4.28 | 0 | 0 | 1 |
| AutoVC[13] | 2.08 | 1.61 | 1.64 | 2.03 | 1.84 | 1.59 | 1.66 | 1.73 | 1.47 | 1.61 | 95.4 | 67.6 | 0.12 |
| VQMIVC[12] | 3.64 | 3.73 | 3.67 | 3.70 | 3.69 | 1.96 | 2.23 | 2.22 | 1.95 | 2.09 | 32.2 | 16.7 | 0.55 |
| NVC-Net[11] | 3.68 | 3.41 | 3.64 | 3.42 | 3.54 | 2.22 | 1.82 | 1.82 | 2.06 | 1.98 | 32.5 | 16.5 | 0.12 |
| PPG-VC[20] | 3.43 | 3.74 | 3.82 | 3.71 | 3.68 | 1.88 | 2.43 | 2.40 | 1.82 | 2.13 | 13.1 | 4.7 | 0.38 |
| HiFi-VC-no-ASR | 2.36 | 2.25 | 2.32 | 2.40 | 2.33 | 3.27 | 3.03 | 2.89 | 3.29 | 3.12 | 91.4 | 62.0 | 0.55 |
| HiFi-VC-no-F0 | **4.37** | **4.43** | **4.45** | **4.46** | **4.43** | **3.55** | 2.83 | 2.77 | **3.36** | **3.13** | **9.2** | 3.7 | 0.33 |
| HiFi-VC | 4.06 | 4.22 | 4.37 | 4.23 | 4.22 | 3.27 | **2.91** | **2.93** | 3.04 | 3.04 | 9.7 | **3.6** | **0.63** |

We use $\lambda_{Rec} = 45$, $\lambda_{AdvP} = 1$, $\lambda_{FM} = 1$, $\lambda_{Spk} = 0.01$. As in traditional GAN optimization, each training step updates the discriminator using the $\mathcal{L}_{AdvD}(s, x)$ loss and other modules with the $\mathcal{L}_P(s, r, x)$ loss.

### 3.3. Implementation Details

During training, all voice samples are converted to 24kHz. ASR produces features with a 40ms period and the F0 predictor generates features every 10ms. The F0 prediction network is designed to downsample F0 features to match those of ASR.

To match the ASR feature frequency, we increase HiFi-GAN upsampling level from 256 to 960. We also remove a bias parameter from the final convolutional layer to stabilize mixed-precision training [27]. We train our network for 120 epochs using Adam optimizer. We set the initial learning rate to 0.0002 and use an exponential scheduler with $\gamma = 0.995$. Each epoch takes 85 minutes on a single NVIDIA V100 GPU.

## 4. Experiments

In this section, we describe the experimental setup and our results, produced by subjective and objective evaluations.

### 4.1. Experimental Setup

We use the VCTK dataset [28] for training baselines and our model. The dataset includes 44242 voice samples from 110 speakers, among which are 47 male, 61 female, and 2 speakers of unknown gender. The distribution of sample lengths is presented in Figure 4. We keep 6 speakers for any-to-any evaluation and use the other speakers during training. Voice samples do not overlap between training and testing.

We compare our method to previous works with publicly available implementations. In particular, we use AutoVC [13], VQMIVC [12] and NVC-Net [11]. The original AutoVC implementation failed to train in our setup, which is why we used an official pretrained AutoVC model for all comparisons. We applied the same AutoVC model to many-to-many and any-to-any tasks. We tried to include StarGANv2-VC [15] in our comparison but encountered several problems. There is a discussion in the original GitHub repository, which highlights the difficulties of training StarGANv2-VC with more than 30 speakers[3]. An-

[3] https://github.com/yl4579/StarGANv2-VC/issues/6#issuecomment-920777258

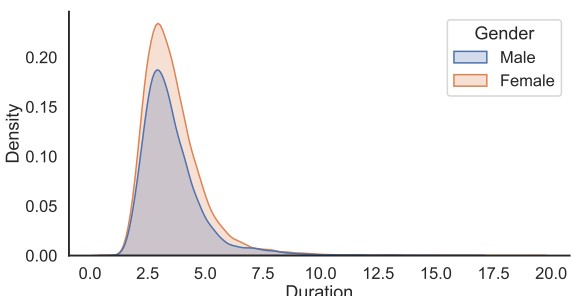

Figure 4: *The distribution of voice sample lengths (seconds) in the VCTK dataset for male and female speakers.*

other difficulty with StarGANv2-VC is that it can't be used for any-to-any conversion. We thus decided to exclude this method from the comparison.

Voice conversion evaluation usually involves the estimation of the output voice quality, voice similarity, linguistic and prosody consistency. We evaluate voice quality using subjective mean opinion score (MOS) [29]. Each predicted record is estimated with a grade ranging from 1 (completely unnatural) to 5 (completely natural).

We use a crowd-sourcing platform for markup. During the markup, we place 11 entries on each page of tasks, including one honeypot, two examples for each possible combination of source and target speaker genders (M2M, F2M, M2F, F2F), as well as two original female and male voice samples. Each algorithm was evaluated by 400 samples in many-to-many setup and by 144 samples in any-to-any. Grades were obtained from 900 assessors with an overlap 10 for many-to-many and 20 for any-to-any.

For voice similarity evaluation, we construct pairs of voice samples. Assessors are required to measure voice similarity between samples with grades between 1 (different) and 5 (same). We report the final MOS for different source and reference gender combinations.

Linguistic consistency is measured as word error rate (WER) and character error rate (CER) between source and output speech samples. As our method involves ASR features, WER and CER can be biased. To solve this problem, we use a different ASR model for evaluation[30]. We also report prosody consistency measured as Pearson correlation coefficient (PCC) between F0 tracks from source and predicted samples.

## 4.2. Conversion Quality

We perform separate evaluations for many-to-many and any-to-any setups. Voice quality and similarity MOS metrics for these two tasks are reported in Table 1 and Table 2 respectively. Among baselines, VQMIVC generally performs better in terms of voice quality, and NVC-Net achieves higher similarity. At the same time, the proposed HiFi-VC method outperforms all considered baselines both in terms of voice quality and similarity.

We also perform a set of objective evaluations aimed at linguistic and prosody consistency. Evaluation results for many-to-many and any-to-any tasks are reported in Table 1 and Table 2 respectively. Among baselines, NVC-Net achieves lower WER and CER in the many-to-many task while being on par with VQMIVC in the any-to-any task. On the other hand, VQMIVC achieves higher pitch consistency compared to other baselines. The proposed HiFi-VC method achieves the lowest WER and CER among all methods. At the same time, the prosody predicted by HiFi-VC better correlates with the source speech sample.

In general, the proposed method achieves high voice quality and similarity. Sometimes it even slightly outperforms ground truth, but the difference isn't significant according to Welch's t-test with a significance level of 5%. At the same time, the superiority to the baselines is significant at the same level.

## 4.3. Ablation Studies

We performed ablation studies for the proposed method. First, we replaced ASR with a neural content encoder, similar to NVC-Net[11]. We also implemented HiFi-VC-no-F0 architecture without an F0 encoder. Evaluation results for these methods are presented in Table 1 and 2. According to these results, the ASR encoder is important for both MOS and objective evaluation metrics. An interesting behavior is observed for the no-F0 method. This method improves most metrics in both many-to-many and any-to-any setups at the cost of F0 quality (PCC). We thus suggest using the method without an F0 encoder in applications, where exact prosody transfer is not required.

## 4.4. Discussion

Experimental results suggest that HiFi-GAN can be used either as a vocoder or as a joint decoder-vocoder module. The proposed HiFi-VC method achieves better conversion results than the baselines in both many-to-many and any-to-any setups. HiFi-VC also reduces the computation cost by excluding separate decoder module from the pipeline. The proposed speaker conditioning scheme increases speaker similarity and the usage of ASR bottleneck and F0 features leads to high linguistic and prosody consistency.

## 5. Conclusion

In this work, we presented a novel voice conversion system, which combines the ASR model with direct waveform prediction using a conditioned HiFi-GAN. A robust ASR feature extractor along with a speaker encoder allows this method to solve general any-to-any conversion tasks. According to multiple experiments involving subjective and objective evaluation, our method achieves better voice conversion than the baselines in terms of voice quality, similarity, and consistency.

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
