# OpenReview forum: "HiFi-VC: High Quality ASR-based Voice Conversion"
_Interspeech.org/2023/Workshop/SSW — SSW12_

### Official Review · Reviewer_cJfd · 2023-06-02
**well written novel paper with HQ results**

**Rating:** 9
**Confidence:** 4

**Review:**

The paper is about any-to-any VC, with highly novel ideas, and having very good results / improvements compared to the state-of-the-art. The sound samples in the supplementary material also support the high-quality results.

Key Strength of the paper
- well written, clear methods, novel idea, well evaluated, very good results

Main Weakness of the paper
- the description of the results / discussion is quite short, despite that it could easily fit to the 6 pages limit. I suggest to extend the analysis of the results


Novelty/Originality, taking into account the relevance of the work for the SSW audience
- the paper is highly novel with the ASR-based idea

Technical Correctness, is the work technically and/or scientifically solid? Are sufficient details provided to allow any experiments to be reproduced or equivalent experiments run?
- the papers is technically correct, with details enough for understading

Suggestions for improvement
- the description of the results / discussion could be extended (Sec. 4.2/4.3/4.4), e.g. statistical significance tests would be beneficial
- Fig. 3: "Speaker embedding is projected using linear layers, repeated along time dimension" - it is not clear from the text, why repeating the speaker embedding is necessary / useful
- Sec 4.2, " HiFi-VC even shows voice quality slightly better than ground truth, but it can be explained by random deviations." - it would be good to see some explanation for this surprising result
- Sec 4.4, "HiFi-VC also reduces the computation cost by excluding separate decoder module from the pipeline." - it would be good to quantify this, i.e. how much is the computational cost lower?

Quality of References, is it a good mix of older and newer papers? Do the authors show a good grasp of the current state of the literature? Do they also cite other papers apart from their own work?
- references are OK

Clarity of Presentation, the English does not need to be flawless, but the text should be understandable
- the paper is well written in general

---

> ### Author Response · Authors · 2023-06-26
> **Answers**
>
> Thanks for taking your time and giving a detailed review regarding the paper.
>
> We performed Welch's t-test with a significance level of 5%. According to the results, the difference between the proposed method and ground truth is not significant. At the same time, the superiority to the baselines is significant at the same level.
>
> We updated the camera-ready version of the paper according to the review.

---

### Official Review · Reviewer_aaKQ · 2023-06-04
**Voice conversion method without intermediate mel generation based conditioning a hiFI-GAN neural vocoder with the ASR output and F0 embeddings of the source audio,  and a VAE embeddings of the target speaker audio.**

**Rating:** 7
**Confidence:** 2

**Review:**

This paper proposes a method for voice conversion that combines an ASR encoder with a hiFi-GAN vocoder without generating any  intermediate mel-spectrogram.  The authors also introduce a VAE speaker encoder with embeddings used to condition each of the layers of the HiFi GAN decoder.

Strong points:
- The method, as an extension of HiFI GAN is relatively simple yet effective .
- Good subjective results

Weakness:

- A strong assumption in the paper is that in voice conversion the prosody (pitch changes and temporal structure) have to be copied from the source speech.  The authors should  make  clearer  for what  applications such prosody transfer is actually required. While most current VC work under the same assumption, this is more due to a limitation of the techniques than to a real use case.  A different speaker saying the same sentence will definitely use different prosody, and that prosody is a strong component of that speaker’s voice identity.   The results actually show that not forcing the source F0 improves the quality.
- Authors do not seem to be very up-to-date with the VC/TTS literature. For example the description of the speaker encoder corresponds 100 to that of a variational auto-encoder (VAE) introduced by Kingma (see https://arxiv.org/abs/1312.6114) which has been widely used to model speakers, styles, etc This should be mentioned, together with the appropriated references.

Other comments and questions:
- Part of the approach is similar to recent papers like VALL-E (https://arxiv.org/abs/2301.02111)  or SPEAR-TTS (https://arxiv.org/abs/2302.03540) . Although a full comparison is obviously well beyond the scope of the paper, some comments related to similarities/differences and potential advantages/disadvantages could be good.
- During training Reference mel and training mel are the same or were the reference  mel samples from the same target speaker but for a different utterance?
-  For the speaker conditioning on the decoder, did you use a different  FC for each layer? Did you try any experiment with  the speaker embedding added only at the input of the joint decoder? For completeness, could you please succinctly describe the internal architecture of the HiFI-GAN decoder layers (whether they are convNet, transformer, etc)?
- Also on the speaker encoder, your approach is very similar to Film (https://arxiv.org/abs/1709.07871) but using only bias vectors and no modulators.
- The title is a little misleading. There are plenty of VC methods based on ASR. Maybe you should try to change (if possible) to something that makes  your approach clearer.

Novelty:  Not outstanding but sufficient.

Technical correctness: it looks correct to me

References: Acceptable but it could benefit from some review of pre-2021 VC literature as the one provided in “An Overview of Voice Conversion and its Challenges: From Statistical Modeling to Deep Learning” (https://www.pure.ed.ac.uk/ws/portalfiles/portal/180614479/SismanEtalIEEE2020AnOverviewOfVoiceConversion.pdf)

Clarity of presentation: Very good

---

> ### Author Response · Authors · 2023-06-26
> **Answers and clarifications**
>
> Thanks for taking your time and giving a detailed review regarding the paper. We hope that the comments below will clarify some questions.
>
> * The voice conversion engine clones the prosody of the input. It differs VC from TTS or the combination of ASR and TTS. TTS can also use an extra voice sample (as in VALL-E) for the target voice parametrization. The technique is sometimes called voice cloning. Both conversion and cloning are valuable tools in applications. Some of them are mentioned in the introduction section of the paper.
> * During training reference and source are equal.
> * We tried to use a single FC layer for all decoder layers, but initial experiments highlighted the superiority of multiple projections. We kept the latter approach for the following experiments.
> * Parametrization is indeed similar to Film. Future work can be focused on additional experiments with conditioning, including scaling vector.
>
> We updated the camera-ready version of the paper according to the review.

---

### Decision · Program_Chairs · 2023-06-14

**Decision:**

Accept

**Comment:**

SSW2003 received 45 papers. The acceptance rate is 82%. We are pleased to inform you that your paper has been accepted by the SSW2023 Program Committee. Please read the reviews carefully and submit your camera-ready paper by June 28th. Most reviewers performed a detailed review. Please answer to their questions and consider their comments. Note that camera-ready papers are credited with one extra page to allow authors to consider reviewers’ suggestions. So max 7 pages in total including figures & refs.
The deadline for submitting the revised version (with full non-anonymized authors and refs!) is 28th June.